# Intestinal Epithelial Barrier Maturation by Enteric Glial Cells Is GDNF-Dependent

**DOI:** 10.3390/ijms22041887

**Published:** 2021-02-14

**Authors:** Michael Meir, Felix Kannapin, Markus Diefenbacher, Yalda Ghoreishi, Catherine Kollmann, Sven Flemming, Christoph-Thomas Germer, Jens Waschke, Patrick Leven, Reiner Schneider, Sven Wehner, Natalie Burkard, Nicolas Schlegel

**Affiliations:** 1Department of General, Visceral, Vascular and Pediatric Surgery University Hospital Würzburg, Oberduerrbacherstrasse 6, 97080 Wuerzburg, Germany; Meir_M@ukw.de (M.M.); felix.kannapin@yahoo.de (F.K.); Ghoreishi_Y@ukw.de (Y.G.); Kollmann_C@ukw.de (C.K.); flemming_s@ukw.de (S.F.); germer_c@ukw.de (C.-T.G.); burkard_n@ukw.de (N.B.); 2Department of Biochemistry and Molecular Biochemistry, University of Wuerzburg, Am Hubland, 97074 Wuerzburg, Germany; markus.diefenbacher@uni-wuerzburg.de; 3Department of Anatomy and Cell Biology University of Munich, Pettenkoferstrasse 11, 80336 Munich, Germany; jens.waschke@med.uni-muenchen.de; 4Department of Surgery, University Clinic Bonn, Venusberg-Campus 1, 53105 Bonn, Germany; patrick.leven@ukbonn.de (P.L.); reiner.schneider@ukbonn.de (R.S.); Sven.Wehner@ukbonn.de (S.W.)

**Keywords:** enteric glial cells, neurotrophic factors, intestinal epithelial barrier, GDNF5, RET6, inflammatory bowel disease, enteric nervous system, gut barrier, intercellular junctions

## Abstract

Enteric glial cells (EGCs) of the enteric nervous system are critically involved in the maintenance of intestinal epithelial barrier function (IEB). The underlying mechanisms remain undefined. Glial cell line-derived neurotrophic factor (GDNF) contributes to IEB maturation and may therefore be the predominant mediator of this process by EGCs. Using GFAP^cre^ x Ai14^floxed^ mice to isolate EGCs by Fluorescence-activated cell sorting (FACS), we confirmed that they synthesize GDNF in vivo as well as in primary cultures demonstrating that EGCs are a rich source of GDNF in vivo and in vitro. Co-culture of EGCs with Caco2 cells resulted in IEB maturation which was abrogated when GDNF was either depleted from EGC supernatants, or knocked down in EGCs or when the GDNF receptor RET was blocked. Further, TNFα-induced loss of IEB function in Caco2 cells and in organoids was attenuated by EGC supernatants or by recombinant GDNF. These barrier-protective effects were blunted when using supernatants from GDNF-deficient EGCs or by RET receptor blockade. Together, our data show that EGCs produce GDNF to maintain IEB function in vitro through the RET receptor.

## 1. Introduction

The intestinal epithelial barrier (IEB) plays an essential role in health and disease. This barrier is maintained by polarized and highly specialized cells, including enterocytes, goblet cells, Paneth cells, enteroendocrine cells and M-cells which line the whole gut lumen [1,2,3]. The intercellular space between these cells is sealed by different junctional proteins such as tight junctions, adherens junctions and desmosomes, which together form the apical junctional complex [3]. Dysregulation of IEB function caused by loss of junctional proteins in the apical junctional complex represents a hallmark in the pathogenesis and perpetuation of inflammatory bowel diseases (IBD) [3,4].

One of the key players critically involved in the regulation of the IEB is the enteric nervous system (ENS) [5], which is built up of enteric neurons and enteric glial cells (EGC). The ENS is organized into two major plexus which are localized in the submucosa and within the muscularis propria. While the effects of enteric neurons in the regulation of the IEB are well documented, the role of EGCs and their predominant effector proteins remain controversial. On the one hand, recent studies using in vivo models in which a non-inflammatory reduction of EGCs was induced [6,7] suggested that enteric glia primarily regulate intestinal motility and secretomotor function, but appeared not to be necessary for the maintenance of IEB function. On the other hand, there is evidence that a toxic or autoimmune ablation of EGCs leads to the breakdown of IEB function followed by a spontaneous jejuno-ileitis comparable to the phenotype seen in Crohn’s disease [8,9]. In line with this, it was shown that EGCs are reduced in the healthy, non-inflamed parts of the intestine of patients suffering from IBD prior to the onset of inflammation. This led to the assumption that EGCs might play an important early role in the pathogenesis of IBD [10,11]. In addition, in vitro co-culture model studies of EGCs and the enterocyte cell line Caco2 demonstrated that EGCs regulate epithelial barrier function and proliferation [12,13]. The effects of EGCs on the IEB may be caused by pro-inflammatory or anti-inflammatory factors which are known to be secreted by EGCs [7]. In this context, EGCs secrete mediators such as nitric oxide and prostaglandin 2 that destabilize the IEB, as well as the neurotrophic factor glial cell line-derived neurotrophic factor (GDNF), 15-deoxy-Δ12,14-prostaglandin J2 and glial-derived S-nitrosoglutathione (GSNO), that all have protective roles on intestinal barrier function through distinct mechanisms [7,13,14].

Previously, we found that GDNF, which is usually secreted in large amounts by EGCs, contributes to intestinal barrier maturation in vitro by regulating the desmosomal cadherin Dsg2 [15]. Further, loss of GDNF strongly correlated with reduced barrier function in patients with IBD. Accordingly, therapeutic administration of GDNF in DSS-induced acute colitis in mice improved intestinal barrier function by upregulating intercellular junctional proteins such as Dsg2 and tight junctions [15].

In view of these findings, we hypothesized that EGCs secrete GDNF at physiologically relevant concentrations and that, among the numerous other factors secreted by EGCs, it might be critical to IEB maturation.

## 2. Results

### 2.1. EGCs Express and Secrete GDNF in Vivo and In Vitro at Significant Levels

To confirm that EGCs express and secrete GDNF at physiologically relevant levels, we used GFAP^Cre^ x Ai14^floxed^ mice which allow visualization of GFAP positive enteric glial cells by tdTomato. Immunostaining showed a co-localization of tdTomato and GFAP at the plexus myentericus in the small intestine of mice (Figure 1A). After sorting of tdTomato positive cells from murine intestines by Fluorescence-activated cell sorting (FACS), *qPCR* confirmed that sorted cells were positive for the glial markers Sox10 and GFAP ex vivo and in vitro in primary EGC cultures (Figure 1B). *QPCR* confirmed that these tdTomato-positive cells express GDNF mRNA (Figure 1C). We compared the expression and secretion of GDNF by EGCs with an established EGC cell line (CRL 2690) [16]. ELISA-based measurements confirmed GDNF levels in human and murine lysates of the terminal ileum at 56.0 ± 4.3 pg/mL or 42.6 ± 7.4 pg/mL, respectively. CRL2690 and primary EGC cultures showed levels of 88.7 ± 5.5 pg/mL and 241.7 ± 44.4 pg/mL in cell culture supernatants, demonstrating that GDNF levels in intestinal tissue lysates and cell culture supernatants range in a comparable level (Figure 1D). Western Blots confirmed that tdTomato-positive cells, as well as CRL2690 cells, express GDNF on the same levels (Figure 1E, Appendix A). These results confirm that EGCs in vitro and in vivo secrete GDNF at significant amounts.

### 2.2. Co-Culture of EGCs and Intestinal Epithelial Cells Leads to Barrier Maturation

To investigate the interaction of EGCs and enterocytes *in vitro,* we established a co-culture system using Caco2 cells and CRL2690 cells. Primary EGCs were not used since they require many animals to generate and vary considerably in yield as well as state [16]. As shown in Figure 1D, CRL2690 cells secrete GDNF at comparable levels to those present in murine and human intestinal lysates making the GDNF levels used in the following experiments physiologically relevant. To detect whether GDNF at these levels sufficiently induces intestinal epithelial barrier maturation, we applied EGC supernatants and recombinant GDNF at doses of 80 pg/mL and 100 ng/mL to immature Caco2 monolayers as we have described previously [15,17]. TER measurements were made concomitant to the application of cell culture supernatants of EGCs and GDNF to Caco2 monolayers. EGC supernatants significantly increased TER values to 3.01 ± 0.1-fold of baseline after 24 h (Figure 2A). Application of recombinant human GDNF to immature Caco2 monolayers resulted in comparable effects on TER values which were augmented to 2.99 ± 0.07-fold of baseline after 24 h at a dose of 100 ng/mL and to 2.52 ± 0.09-fold of baseline at a dose of 80 pg/mL without significant differences between the experimental conditions (Figure 2A). Western blots confirmed the presence of the GDNF receptors GFRα1-3 as well as the RET receptor in Caco2 cells (Figure 2B, Appendix A).

Next, we established the same co-culture system of EGCs (CRL2690 cells) and enterocytes (Caco2 cells) on a transwell filter system, in which EGCs were seeded onto the basal compartment and Caco2 cells were grown on the transwell insert (Figure 2C). Measurements of 4 kDa FITC-Dextran flux were then carried out in this model to estimate the permeability coefficient (P_E_) under these conditions after 24 h of co-culture. The P_E_ provides an objective measure for diffusive permeability across epithelial monolayers as a function of diffusion area, time and change of fluorescence intensities from the apical to the basolateral compartment [18]. Compared to immature control Caco2 monolayers, which displayed a basal P_E_ of 1.19 ± 0.09 cm/s × 10^−6^, the co-culture of EGCs with Caco2 cells had a 20% decrease of P_E_ to 0.99 ± 0.07 cm/s × 10^−6^ (Figure 2D). Immunostaining of immature Caco2 monolayers grown on filters showed an irregular pattern of the desmosomal protein Desmoglein2 (Dsg2), the adherens junction protein E-Cadherin and the tight junction protein Claudin1 at the cell borders under control conditions (Figure 2E). This was considerably changed in the presence of EGCs when these junctional proteins were found to be augmented at the borders and showed a regular staining pattern (Figure 2E).

### 2.3. Depletion of GDNF from EGC Supernatants Blunts Intestinal Barrier Maturation

To dissect the specific contribution of GDNF on intestinal barrier maturation from other mediators in supernatants from EGCs, we depleted GDNF from EGC supernatants using sepharose beads coated with anti-GDNF antibodies. This was verified by western blotting where the GDNF-specific band at 15 kDa was hardly detectable in GDNF-depleted supernatants whereas a strong band for GDNF was observed in supernatants that had not been depleted of GDNF (Figure 3A–C). Loading of the Western blot membrane with GDNF/GDNF-antibody/sepharose beads fraction centrifuged from the EGC supernatant resulted in the detection of the GDNF-specific band. ELISA-based measurements of GDNF levels in EGC supernatants after depletion of GDNF showed a reduction of 83 ± 10% compared to GDNF from control supernatants (Figure 3C). Application of GDNF-depleted supernatants to immature Caco2 monolayers showed no changes of TER in the time course of measurements compared to the significant increase following the application of non-depleted supernatants (Figure 3D), indicating the requirement of GDNF for epithelial barrier maturation. This set of experiments points to a specific role for EGC-derived GDNF in comparison to other mediators that were still present in the EGC culture supernatant.

### 2.4. Knockdown of GDNF in EGCs Attenuates Intestinal Epithelial Barrier Maturation

A stable GDNF knockdown cell line from EGCs (EGC^GDNF KD^) was generated using the CRISPR/Cas system. The knockdown of GDNF resulted in a reduction to 50 ± 6% of GDNF levels as revealed by Western blot analyses (Figure 4A,C) from cell culture lysates. In cell culture supernatants, Western blots (Figure 4B,D) showed a reduction of GDNF secretion of 80% of controls in EGC^GDNF KD^. EGC supernatants from EGC and from EGC^GDNF KD^ were collected and incubated with Caco2 cells. In TER measurements of Caco2 monolayers treated with supernatants from EGC cells, TER values were augmented significantly to 3.2 ± 0.12-fold of baseline after 24 h (Figure 4E). In contrast, supernatants from EGC^GDNF KD^ led to a significantly less pronounced increase of TER to 1.99 ± 0.06-fold of baseline compared to untreated controls after 24 h. This confirmed that GDNF is required for EGC-mediated barrier maturation. The fact that GDNF was still present in EGC^GDNF KD^ which still exerted barrier-protective effects on Caco2 monolayers serves to confirm a dose-dependent effect of GDNF in epithelial barrier maturation.

### 2.5. Effects of EGCs on the Distribution of Junctional Proteins at the Cell Borders Are GDNF-Dependent

To visualize the effects of EGCs and GDNF on junctional proteins in confluent immature Caco2 monolayers, immunostaining for the desmosomal protein Dsg2, tight junction proteins Claudin5 and Claudin1 and adherens junction protein E-Cadherin was carried out. Both the application of cell culture supernatant from EGCs (Figure 5i–l) and application of recombinant GDNF (Figure 5e–h) augmented junctional staining of Dsg2, Claudin5 and 1 and E-Cadherin when compared to Caco2 controls (Figure 5a–d). This qualitative evaluation was confirmed quantitatively as EGC supernatant increased the junctional staining of Claudin1 to 1.81 ± 0.11- fold of controls, Claudin5 to 2.02 ± 0.19- fold of controls and Dsg2 to 1.55 ± 0.1- fold of controls. Similarly, application of recombinant GDNF enhanced the staining of Claudin1 to 1.62 ± 0.12-fold of controls, Claudin5 to 1.25 ± 0.08- fold of controls and Dsg2 to 1.44 ± 0.17-fold of controls. E-cadherin however was unchanged in the quantifications (Appendix A). This was not observed when monolayers were incubated with cell culture supernatant from EGC^GDNF KD^ (Figure 5m–p) and with GDNF-depleted supernatants from EGCs (Figure 5q–t; Appendix A). Western blots revealed that the overall expression of junctional proteins was not changed following the treatment with EGC supernatant, recombinant GDNF or supernatant from EGC^GDNF KD^ (Appendix A). Quantification of western blots showed that Dsg2 protein levels were not changed following GDNF treatment (β-Actin correlated optical density of 0.80 ± 0.03 compared to 0.82 ± 0.04 under control conditions). Similarly, in co-cultures of Caco2 cells with EGC compared to Caco2/EGC^GDNF KD^ co-culture did not change Dsg2 levels (optical density 0.85 ± 0.07 compared to 0.84 ± 0.03). Tight junction protein levels of Claudin 1 (99 ± 4% of controls) and Claudin5 (99 ± 6% of controls) were also unchanged following application of GDNF as well as in co-cultures of Caco2 cells with EGCs or EGC^GDNF KD^, respectively (Appendix A). This is in line with our previous findings [15,17] where we observed that GDNF has no effects on the expression of junctional proteins but recruits them to the cell borders in terms of barrier-maturation.

### 2.6. EGC-Mediated Epithelial Barrier Stabilization Is Dependent on the GDNF Receptor RET

Previously, we demonstrated that GDNF-induced effects on enterocytes were dependent on the presence of the RET receptor [15]. To provide further evidence for a specific role of GDNF secreted by EGCs on Caco2 monolayers, we used the specific RET inhibitor Blue667 [19]. As revealed by Western blotting, application of GDNF led to an increase of phosphorylation of RET at site Y1062 to 1.31 ± 0.09- fold of controls (Figure 6A). Simultaneous incubation of Caco2 monolayers with GDNF together with Blue667 blocked RET phosphorylation to 67.5 ± 9% of untreated controls (Figure 6A,B). This confirmed that RET signaling is critically involved in GDNF signaling in enterocytes. Measurements of TER demonstrated that application of GDNF and EGC supernatant both increased the TER to 1.90 ± 0.17-fold and to 1.83 ± 0.06-fold of baseline, respectively after 24h whereas this TER increase was significantly less pronounced under control conditions. Incubation of Caco2 cells with Blue667 attenuated GDNF-induced and EGC supernatant-induced increase of TER (Figure 6C,D). TER measurements following Blue667 alone and in combination with GDNF or EGC supernatant did not show significant differences.

### 2.7. The Effects of EGC Co-Culture on the Inflammation-Induced Breakdown of Intestinal Epithelial Barrier Function Are GDNF-Dependent

Previous studies demonstrated that GDNF release is altered in inflammation [20,21]. To gain further insight into the role of EGCs on barrier dysregulation in the context of inflammation, we evaluated the effects of pro-inflammatory mediators on EGCs. To this end, EGCs were incubated with tumor necrosis factor α (TNFα), lipopolysaccharide (LPS) and Lipoteichoic acid (LTA), which activates Toll-like receptor2 (TLR2). Interestingly, incubation of EGC with TNFα and LPS but not LTA increased expression of GDNF as revealed by Western blot analyses of cell lysates (Figure 7A,B). The release of GDNF was significantly augmented following incubation of EGCs with TNFα and LTA but not after application of LPS (Figure 7C,D). Control experiments to exclude GDNF release due to cell death demonstrated that EGC cell viability as revealed by MTT assays was unaltered following application of TNFα, LPS or LTA whereas the apoptosis inductor staurosporine led to increased cell death (Figure 7E).

Next, we tested whether the increased expression and secretion of GDNF induced by pro-inflammatory mediators would be sufficient to protect Caco2 from TNFα-induced loss of barrier function. Incubation of Caco2 monolayers with TNFα alone resulted in a significant increase of 4 kDa FITC Dextran flux in the transwell model, where P_E_ increased to 1.78 × 10^−6^ ± 0.10 × 10^−6^ compared to 1.32 × 10^−6^ ± 0.08 × 10^−6^ under control conditions (Figure 8A). Co-culture of Caco2 cells with EGC inhibited TNFα-induced loss of barrier function. Application of RET inhibitor Blue667 to TNFα-challenged Caco2/EGC co-culture demonstrated that the barrier protective effect of EGCs was blunted (Figure 8A). In line with these observations, co-culture of EGC^GDNF KD^ did not protect Caco2 monolayers from TNFα-induced loss of barrier properties (Figure 8B).

Finally, immunostaining revealed the junctional proteins Dsg2, Claudin1, 5 and E-Cadherin to be distributed regularly at the borders in differentiated Caco2 cells under control conditions (Figure 8C(a–d)). The application of TNFα led to a reduction of all these proteins at the cell borders (Figure 8C(e–h)). Quantification of immunostaining confirmed Claudin1 to be reduced to 56 ± 3%, Claudin5 to 72 ± 6% and Dsg2 to 64 ± 5% of controls (Appendix A). Application of EGC supernatants to Caco2 monolayers EGCs blocked TNFα- induced delocalization of junctional proteins (Figure 8C(i–l) and Appendix A)). This effect was abolished when EGC supernatant together with RET inhibitor Blue667 was incubated with TNFα on Caco2 monolayers (Figure 8C(m–p); Appendix A).

### 2.8. EGC-Mediated Effects on IEB Function on Junctional Proteins in Intestinal Organoids Are GDNF-Dependent

To confirm the observation of a specific role for GDNF-dependent effects mediated by EGCs, we used murine organoids as primary intestinal epithelial cells. Functional measurements were carried out by assessing the ratio of 4 kDa FITC-dextran translocation across epithelial organoids as described previously [22]. Application of TNFα to mimic inflammation-induced changes of barrier function led to a significant translocation of 4 kDa FITC Dextran across organoids (Figure 9A,B). This was blocked when recombinant GDNF or supernatant from EGCs were applied to the organoid cultures. Incubation of enteroids with RET-inhibitor Blue667 blunted barrier-protective effects of GDNF and EGC supernatants which verified a predominant effect of GDNF (Figure 9A,B). These observations were paralleled by changes of junctional proteins comparable to those seen in Caco2 cells. Immunostaining of organoids revealed that Dsg2, Claudin1, 5 and E-Cadherin were all regularly distributed at the cell borders (Figure 9C(a,g,m,s); Appendix A). Application of TNFα led to a reduction of all proteins at the cell borders like the effects observed in Caco2 cells (Figure 9C(b,h,n,t); Appendix A). Application of GDNF (Figure 9C(c,i,o,u) or EGC supernatants (Figure 9C(e,k,q,w); Appendix A) inhibited TNFα-mediated effects on the junctional proteins. These beneficial effects of GDNF and EGC supernatants on junctional proteins were abolished by RET inhibitor blue667 (Figure 9C(f,l,r,x); Appendix A).

## 3. Discussion

The present study extends our previous investigations on the role of neurotrophic factor GDNF in the regulation of the intestinal epithelial barrier. Previously, we demonstrated that GDNF induces barrier maturation of intestinal epithelial cells and protects from loss of intestinal barrier function in inflammation that plays a key role in patients with IBD [15,17,23]. Here, we focused on the crosstalk between epithelial cells and EGCs that, among various other factors, secrete significant amounts of GDNF and may potentially contribute to the regulation of the intestinal barrier. We demonstrated that EGCs in vivo and in vitro secrete GDNF at physiologically relevant doses as can be found in murine or human intestine. These doses are effective to induce barrier maturation and inhibit inflammation-induced breakdown of the intestinal epithelial barrier. To test for the specific role of GDNF in this context, we performed a series of in vitro co-culture experiments. We show here that both depletion of GDNF from EGC supernatants, as well as knock-down of GDNF in EGCs, blunted the effects on barrier maturation of GDNF in Caco2 monolayers. This demonstrates a predominant role for GDNF in this context.

### 3.1. Considerations of the Co-Culture System of EGCs and Caco2 Cells

Testing our hypothesis that neurotrophic factor GDNF is a key player in EGC-mediated intestinal barrier regulation required a valid, stable and reproducible in vitro model. Primary murine isolated EGCs, primary cultivated EGCs and immortalized CRL2690 EGCs all express and secrete significant amounts of GDNF. The observation that GDNF levels in primary EGCs are significantly higher than the levels of CRL2690 in ELISA-based measurements should be interpreted with care since these measurements were not adjusted to the absolute amount of cells in the cell culture flasks from which the cell culture supernatants were collected. This is supported by Western blot analyses where the expression of GDNF in both cell lines did not differ. The use of intestinal tissue lysates to quantify GDNF levels and the comparison with cell culture supernatants served to verify that the experimental conditions chosen for the present study were in a physiological range. The results obtained from these measurements justify the argument that the coculture experiments with CRL2690 cells secreting levels of GDNF in the range of 50–100 pg/mL reflect the overall levels of GDNF observed in the in vivo situation [16]. Further, this model offers important technical advantages such as easy handling, high reproducibility of conditions and especially the use of a standard cell culture medium as opposed to the use of primary EGC cultures. These are often difficult to handle and the findings from co-culture models with primary cells are often difficult to reproduce. Besides, GDNF is highly conserved throughout different species, which is exemplified by the fact that rat and human GDNF share more than 90% amino acid sequence identity as revealed by comparison in databases [24]. Importantly, previous investigators have cultivated and compared the properties of EGCs from different species including rats, mice and humans. They demonstrated that EGCs have similar functional properties on the intestinal barrier independently of the species [24]. Therefore, it appeared reasonable for us to combine the immortalized EGC cell line derived from the rat intestine [16] for our co-culture system together with human Caco2 cells. Caco2 cells derive from a human colon carcinoma [25] and are a widely used in vitro model to investigate intestinal barrier function, since they share many aspects of enterocyte morphology and function [26]. In summary, the co-culture model used in the present study shows stable and highly reproducible conditions that appeared to be ideal for us to investigate the specific role of GDNF in the context of EGC-mediated IEB regulation. In addition, we used murine intestinal organoids to verify the key findings of EGC-mediated effects in primary intestinal epithelial cells.

### 3.2. EGC-Mediated Effects on Intestinal Epithelial Barrier Function

In our experiments, the co-culture of EGCs with enterocytes induced a faster differentiation of barrier properties of Caco2 monolayers as revealed by measurements of TER and 4 kDa FITC dextran flux. The barrier-stabilizing effects mediated by the presence of EGCs were consistent with augmented staining patterns of critical junctional proteins such as Dsg2, E-Cadherin and Claudin1, 5 at the cell borders. As revealed by Western blotting GDNF did not augment the expression of junctional proteins. This is supported by our own previous findings where GDNF induced barrier-maturing effects by inducing the recruitment of junctional proteins to the cell borders [17]. The co-culture experiments confirmed that the presence of EGCs critically contributes to barrier maturation of intestinal epithelial cells which is supported by previous literature [5,12,13,21,27]. In addition, experiments using primary intestinal epithelial cells confirmed the barrier-protective effects of EGC-supernatants after they were stimulated with TNFα. Nevertheless, it must be emphasized that this view has been challenged by in vivo approaches in which loss of glial activity even had beneficial effects on intestinal barrier properties [28]. Moreover, genetic ablation of EGCs did not affect IEB function [6] and germ-free mice appeared to have normal transepithelial resistance even though mucosal EGCs were not present in these animals [29]. While compensatory mechanisms may have considerably influenced the latter observations, the reasons for these controversial findings are not fully understood and will require more detailed in vivo studies on the interactions between EGCs and intestinal epithelial cells [7]. Nonetheless, our reductive in vitro co-culture system presented here strongly supports the studies favoring the important contribution of EGCs to IEB regulation.

### 3.3. GDNF Is a Key Mediator of Barrier Stabilization by EGCs In Vitro

Previously, many different factors such as 15-desoxy-delta-prostaglandin J2 [14], 15-HETE [11] and proEGF [30] secreted by EGCs have been proposed to mediate their effects on intestinal barrier function. However, the specific contribution of GDNF on the effects in co-culture models of EGCs with enterocytes has not been addressed before. In the present study, loss of GDNF was induced by its depletion from supernatants or its knockdown in EGCs. Both conditions clearly attenuated the beneficial effects on barrier functions of Caco2 cells which confirmed the predominant role for GDNF to mediate protective effects on IEB function. Although the incubation of EGC supernatants showed some additional effects compared to the application of recombinant GDNF, it can be concluded that the main effects of EGCs were mediated by GDNF. Further, inhibition of the RET receptor, which we previously confirmed to be critical for GDNF-mediated signaling in enterocytes [15], blocked EGC-mediated barrier stabilization in Caco2 cells. The specific RET-inhibitor Blue667 or Pralsetinib is also used in clinical trials to treat RET-driven endocrine tumors. It is interesting to note that in line with our observations of GDNF-mediated effects on enterocytes, the first clinical trials reported gastrointestinal side effects such as diarrhea or constipation following treatment with Blue667 in up to 16% of all patients [19,31]. Currently, the detailed downstream mechanisms by which RET-signaling affects junctional proteins in intestinal epithelial cells are not characterized. However, it is known that signaling pathways involved in the regulation of junctional proteins such as mitogen-activated kinase pathway (MAPK) may be affected by RET as revealed by studies using the RET inhibitor Vandetanib [32]. Comparable changes of MAPK signaling have been observed in IEB in patients suffering from IBD [15,17,33]. The detailed intracellular mechanisms of RET-driven intestinal barrier maturation will have to be evaluated in future studies.

When considering other factors secreted by EGCs that have previously been implicated in IEB stabilization, there are important differences to consider: For 15-d-PDG-J2, it was shown that it affects epithelial cell proliferation and differentiation through the PPAR_Ɣ_ pathway. This led to the conclusion that this may affect intestinal barrier function although this has not been specifically tested e.g., by using permeability measurements [14]. Similarly, proEGF exerts its barrier-protective function primarily in the context of epithelial wound closure [30]. Another study focused on 15-HETE which was shown to significantly enhance intestinal barrier function by increasing tight-junction-associated protein ZO1 expression [11]. The main discrepancy with our present study is that the authors applied conditioned media to induce 15-HETE expression in rat and human EGCs whereas normal DMEM culture media, as used in our study, hardly expressed 15-HETE. Therefore, it is conceivable that the amount of 15-HETE in our experimental setup is negligible.

When considering the physiological significance of our observations, it is noteworthy that not only EGCs, but also smooth muscle cells [34] and enterocytes [17,23], express and secrete GDNF. In our co-culture system, the relative contribution of enterocyte-derived GDNF appears to be negligible since the depletion of GDNF from EGC supernatants followed by application on Caco2 monolayers or intestinal organoids attenuated all EGC-mediated effects on epithelial barrier function. The relative contribution of smooth muscle cell-derived GDNF, however, remains unclear at present. Nonetheless, it can be speculated whether this additional source of GDNF may be able to compensate for conditions when EGCs are lost. This in turn may explain the above-mentioned conflicting data why the loss of EGCs did not affect IEB function significantly in some models [6].

### 3.4. GDNF Secretion from EGCs Is Stimulated by Inflammatory Mediators Leading to Epithelial Barrier Protection

Pro-inflammatory mediators such as TNFα, LPS or LTA increase the expression of GDNF in EGCs [35]. This was also observed in our experiments where the application of TNFα and LTA on EGCs increased GDNF levels in cell culture supernatants. Importantly, the release of GDNF was not related to augmented cell death by TNFα and LTA as revealed by cell viability assays. Rather, in our co-culture model, TNFα-induced increase of epithelial permeability was blunted in the presence of EGC and TNFα. Again, the protective effects of EGCs on inflammation-induced barrier disruption proved to be clearly GDNF-dependent since no barrier protection was evident in co-cultures with EGC^GDNF KD^ or after application of the RET inhibitor Blue667. A comparable observation was made using intestinal organoids in which both barrier protective effects of EGC supernatants and GDNF after stimulation with TNFα were blunted when the GDNF receptor RET was blocked. These in vitro findings support the view that the release of GDNF by EGCs may be part of a rescue mechanism to protect IEB function in acute inflammation [21]. However, it can be assumed that this rescue mechanism is restricted to the acute onset of inflammation only since it was observed that EGCs are reduced in chronic inflammation in IBD patients [8,10,36]. This is in line with our previous observation where we found a strong reduction of intestinal GDNF levels in areas of severe chronic inflammation in IBD patients [15].

Taken together, our in vitro data demonstrate a significant effect of EGCs on epithelial barrier maturation under basal conditions and following application of TNFα. Knock-down experiments, depletion of GDNF from EGC culture supernatants as well as specific inhibition of the GDNF receptor RET blunted EGC-mediated effects on barrier function in Caco2 monolayers. This confirms our hypothesis that GDNF is a key player in the interaction between EGCs and intestinal epithelial cells. The relevance for this observation in a living organism will have to be verified in detail in appropriate in vivo models in the future.

## 4. Materials and Methods

### 4.1. Cell-Culture

As described previously, Caco2 cells were used as a model of intestinal epithelial cells [18,26] and CRL2690 as a cell line for enteric glial cells [16]. Caco2 cells and CRL260 cells were acquired from ATCC (Wesel, Germany) and grown at 37 °C in a 5% CO_2_ humidified atmosphere. DMEM (Sigma-Aldrich, St. Louis, USA) supplemented with 50 U/mL Penicillin-G, 50 µg Streptomycin and 10% fetal calf serum (FCS, Biochrom, Berlin, Germany) was used for cultivation. Cultures were used for experiments when grown to confluent monolayers. For all experiments, cells were serum-starved for 24 h.

### 4.2. Animals and Intestinal Organoid Generation

For murine organoids experiments were performed on male C57BL/6J mice (Janvier Labs). The studies were approved on 17.07.2019 by the Governments of Unterfranken and Germany (Identification code 2-852). Animals were kept under conditions that complied with the NIH Guide for the Care and Use of Laboratory Animals. Animals were kept on a standard diet and 12-h day and night cycles. To generate organoids, intestinal epithelial cells were isolated from mice intestine as described previously [37].

### 4.3. Organoid Generation

For the organoid generation, the mucosa was mechanically dissected from the submucosa and then villi were removed using a sterile glass slide. The remaining tissue was transferred into a 50 mL falcon tube with 20 mL 4 °C cold hanks balanced salt solution (HBSS) (Sigma-Aldrich, St. Louis, MO, USA), vortexed for 5 s and the supernatant discarded. This washing step was repeated until the supernatant was entirely cleared of cell debris. Afterwards, the tissue was incubated in 4 °C Ethylenediaminetetraacetic acid (2 mM) solved in HBSS (Sigma- Aldrich, St. Louis, MO, USA) for 30 min under gentle rotation on a shaker. Subsequently, the tissue was washed manually in 20 mL HBSS by inverting the tube five times. The mucosa was then transferred into a new tube with 10 mL HBSS and shaken five times manually. This shaking procedure was repeated four times always using a new tube. The amount and size of crypts within drops was checked under the microscope. The supernatants containing the most vital appearing crypts were pooled and centrifuged at 350 g for 3 min at room temperature (RT). Then, the pellet was resuspended in 10 mL basal medium, Dulbecco’s Modified Eagle’s Medium (DMEM)-F12 Advanced (Invitrogen, Carlsbad, CA, USA) supplemented with 1× N2, 2× B27, 1× Anti-Anti, 10 mM 4-(2-hydroxyethyl)-1-piperazineethanesulfonic acid (HEPES), 2 mM GlutaMAX-I (all from Invitrogen, Carlsbad, CA, USA), 1 mM N-acetylcysteine (Sigma-Aldrich, St. Louis, MO, USA), and the crypt number was estimated in a 10 µL drop by microscopic observation. Crypts were centrifuged 1.5 mL tube at 350 g for 3 min at room temperature and the supernatant was removed. The tube with the cell pellet was placed on ice until it was resuspended in cold Matrigel (Corning, Hickory, NC, USA) with 5000 crypts/mL. Drops of 50µL per well were seeded in a 24- well plate and incubated for 10–20 min until the Matrigel was solidified. The culture medium contained a mixture of 50% fresh basal medium and 50% Wnt3A-conditioned medium. Furthermore, the following growth factors were added: 500 ng/mL hR-Spondin 1, 50 ng/mL human Epidermal growth factor (hEGF) (both PeproTech, Rocky Hill, NY, USA), 100 ng/mL rec Noggin (PeproTech, Rocky Hill, NY, USA), 10 mM Nicotinamid, 10 µM SB202190, 10 nM [Leu15]-Gastrin I (all three Sigma-Aldrich, St. Louis, MO, USA), 500 nM A83-01 (Tocris Bioscience, Bristol, UK) and 500 nM LY2157299 (Axon MedChem, Reston, VA, USA). Additionally, 10 µM Rho-kinase inhibitor Y-27632 (Tocris Bioscience, Bristol, UK) was added after seeding and after each splitting. The cells were used as organoid and enteroid culture for 8–10 weeks.

To generate murine EGCs, experiments were performed using 8–12-week-old mice kept in a pathogen-free animal facility with standard rodent food and tap water ad libitum. GFAP^cre^ x Ai14^floxed^ mice were used for the FACS experiments to sort tdTomato positive and *Hoechst* negative EGCs for gene expression analysis.

### 4.4. Generation and Cultivation of Primary EGC

Primary enteric glia cell cultures were obtained by sacrificing C57BL/6 mice 8–16 weeks of age, extracting the small intestine as described previously [38]. After cleansing it with 20 mL of oxygenated Krebs-Henseleit buffer (126 mM NaCl; 2.5 mM KCl: 25 mM NaHCO_3_; 1.2 mM NaH2PO4; 1.2 mM MgCl_2_; 2.5 mM CaCl_2_, 100 IU/mL Pen, 100 IU/mL Strep and 2.5μg/mL Amphotericin), the small bowel was cut in 3–5 cm long segments and kept in oxygenated ice-cold Krebs-Henseleit buffer. Each segment was then drawn onto a sterile glass pipette and the muscularis externa (ME) was stripped with forceps to collect muscle tissue for further digestion steps. After centrifugation at 300 g for 5 min, the tissue was incubated for 15 min in 5 ml DMEM containing Protease Type1 (0.25 mg/mL, Sigma-Aldrich, St. Louis, MO, USA) and Collagenase A (1 mg/mL, Sigma-Aldrich, St. Louis, MO, USA) in a water bath at 37 °C, 150 rpm. The enzymatic digestion was stopped by adding 5 mL DMEM containing 10% FBS (Sigma-Aldrich, St. Louis, MO, USA), centrifugation for 5 min at 300 g and resuspended in proliferation medium (neurobasal medium with 100 IU/Pen, 100 μg/mL Strep, 2,5μg/mL Amphotericin (all Thermo Scientific , Waltham, MA, USA), FGF and EGF (both 20 ng/mL, Immunotools). Cells in proliferation media were kept at 37 °C, 5% CO2 for 4 days to enhance the formation of enteric neurospheres. These enteric neurospheres were dissociated with trypsin (0.25%, Thermo Scientific) for 5 min at 37 °C for experiments and distributed at 50% confluency on Poly-Ornithin (Sigma-Aldrich, St. Louis, MO, USA) coated 6 well plates in differentiation medium (neurobasal medium with 100IU/Pen, 100 μg/mL Strep, 2,5 μg/mL Amphotericin, B27, N2 (all Thermo Scientific, Waltham, MA, USA ) and EGF (2 ng/mL, Immunotools, Friesoythe, Germany). After 7 days in this differentiation medium, mature enteric glia cells and the conditioned medium could be used for FACS, ELISA and Western Blotting.

### 4.5. Gene Expression Analysis of Tdtomato Positive EGCs

Primary EGC cultures, generated from GFAP^cre^ × Ai14^floxed^ mice or fresh digested ME tissue from small bowel isolated from GFAP^cre^ × Ai14^floxed^ mice, was used to sort Td-Tomato negative or positive living cells by FACS using a BD FACS Aria III. Cell viability was confirmed by Hoechst staining and only Hoechst negative cells were sorted. All cells were collected, and RNA isolation was performed by the RNeasy Mini Kit (Qiagen, Hilden, Germany). Complementary DNA was synthesized using the High-Capacity cDNA Reverse Transcription Kit (Applied Biosystems, Darmstadt, Germany) and mRNA expression was quantified by real-time *RT-PCR* using the primers listed in Table 1.

### 4.6. Cell Culture Supernatant

Enteric Glia Cells (CRL2690) were grown to confluence in 6-well plates. After rinsing with PBS, cells were incubated with serum-starved medium for 24 h. Supernatants were then centrifuged at 12.000 g for one minute to remove the remaining cells. For Western blot and ELISA measurements, the supernatant was subsequently concentrated with 10 kDa Amicon Ultra-0.5 Centrifugal Filter Units (Merck Millipore, Darmstadt, Germany), as specified by the manufacturer and Halt protease inhibitor cocktail was added at a 1:100 dilution (Thermo Fisher Scientific, Waltham, MA, USA).

### 4.7. Human Tissue Samples

The Ethical Board of the University of Wuerzburg approved the use of human tissue samples on 28.08.2017 for this study (proposal numbers 113/13, 46/11, 42/16). The specimens were collected from patients that had an indication for surgery. All tissue samples were derived from the terminal ileum. All patients gave their informed consent prior to surgery. All methods were carried out in accordance with relevant guidelines and regulations including the declaration of Helsinki as well as German and European law.

### 4.8. Enteroid Permeability Assay

To evaluate the paracellular permeability in enteroids we used a modified assay previously described by Bradenbacher et. al. [22]. Enteroids were plated in a 48-well chamber (VWR, Darmstadt Germany). After treatment for 24 h, as indicated in Figure 9, enteroids were rinsed three times with PBS and incubated with medium without phenol red (Sigma-Aldrich,St. Louis, MO, USA) containing 10 mg/mL 4 kDa FITC- Dextran and additional test reagents and put on a shaker at 37 °C. After one hour, the wells were washed with PBS for three times and then pictures of each enteroid was taken using a fluorescence microscope (Keyence, Osaka, Japan). Then, mean total fluorescence intensities (average of three representative areas) outside and inside the enteroid was measured using ImageJ. The quotient of the mean total fluorescence from the inside of the enteroid to the outside of the enteroid served as a readout for intestinal permeability.

### 4.9. Test Reagents

Caco2 cells were treated with recombinant human GDNF (PeproTech, Hamburg, Germany). Preliminary dose-response experiments confirmed that GDNF was effective at a dose of 100 ng/mL in Caco2 cells [15,17]. TNFα (Biomol GmbH, Hamburg, Germany) and LPS (Sigma-Aldrich, St. Louis, USA) were both used at 100 ng/mL as previously described [15]. The TLR2-agonist staphylococcal lipoteichoic acid (LTA) (InvivoGen, San Diego, CA, USA) was used at a concentration of 500 ng/mL as suggested by the manufacturer and in previous publications [39,40]. The RET-inhibitor Blue667 (Pralsetinib) (MedChemExpress, Monmuth Junction, NJ, USA) specifically inhibits the phosphorylation of wild-type and oncogenic RET variants [19]. Preliminary dose-response experiments proved effective inhibition of the upstream RET phosphorylation site RET^Y1062^ at a concentration of 1 µM.

### 4.10. Western Blot

For Western blot analysis, cells were grown on 6-well plates. The cells were scratched off the plates and then homogenized in SDS lysis buffer containing 25 mmol/L HEPES, 2 mmol/L EDTA, 25 mmol/L NaF and 1% SDS. SDS gel electrophoresis and blotting were carried out after normalization of protein amount using the BCA assay (Thermo Fisher, Waltham, MA, USA) as previously described [41,42]. Mouse-anti GDNF (R + D Systems, Abingdon, UK) was used at a dilution of 1:150 in 5% bovine serum albumin (BSA) and 0.1% Tween. Mouse antibodies against GFRα1-3 (all Abcam, Cambridge, UK), mouse anti-RET (Abcam, Cambridge, UK) and mouse anti-Phospho-RET^Y1062^ (R&D Systems, Minneapolis, USA) were used at a dilution of 1:600. Horseradish peroxidase-labeled goat anti-rabbit IgG and goat anti-mouse IgG (all Santa Cruz Biotechnology, Heidelberg, Germany) were used (1:3000 in 5% BSA, 0.1% Tween) as secondary antibodies. Peroxidase-labeled β-Actin antibody (Sigma-Aldrich, Munich, Germany) was used to validate equal protein loading. In the absence of a consistently expressed protein marker for cell supernatants, loading of the SDS gel was controlled by volume. Chemiluminescence signal detection and quantifications were performed by densitometry (ChemicDoc Touch Bio-Rad Laboratories GmbH, Munich, Germany).

### 4.11. Immunocytochemistry

Cultured epithelial cells and primary EGCs were grown to confluence on coverslips or transwell filter chambers (0.4µm pore size, Falcon, Heidelberg, Germany). After incubation with or without different mediators or supernatants for 24 h, cells were fixed with 2% formaldehyde for 10 min and then permeabilized with 0.1% Triton-X for 15 min at room temperature. Monolayers were next incubated at 4 °C overnight with mouse anti-Desmoglein2 (Thermo Fisher Scientific, Waltham, MA, USA) and mouse anti-E-Cadherin (BD Biosciences, Franklin Lakes, USA) diluted 1:100 with phosphate-buffered saline (PBS) or rabbit anti-Claudin1 and rabbit anti-Claudin5 (Both Thermo Fisher Scientific, Waltham, MA, USA) diluted 1:50 with PBS, as well as chicken anti GFAP (Abcam, Cambridge, UK) diluted 1:60 in PBS. We used Cy3- or 488- labeled goat anti-mouse or goat anti-rabbit (all diluted 1:600, Dianova, Hamburg, Germany) and donkey anti-chicken FITC (Biozol, Eching, Germany as secondary antibodies. Coverslips and filters were placed on glass slides. To counterstain cell nuclei Vector Shield Mounting Medium as an anti-fading compound, which included DAPI (Vector Laboratories, Burlingham, CA, USA) was used. Representative experiments were photographed with a confocal microscope (Leica TCS SP2, Wetzlar, Germany).

### 4.12. qRT-PCR

RNA from EGC wildtype and GDNF knock-out cells was isolated using *TRIZOL* and cDNA was synthesized with an iScript cDNA Synthesis Kit (Biorad, Munich, Germany). qRT-PCR was performed using MESA GREEN qPCR MasterMix Plus for SYBR Assay No ROX (Eurogentec, Cologne, Germany) on the CFX96 Touch Real-Time PCR Detection System (Biorad, Munich, Germany). Gene expression was analyzed via the Bio-Rad CFX Manager software with beta-2-microglobulin (B2M) as a reference gene. All reactions were performed in duplicates at 60.0 °C annealing temperature. Primers were applied at a concentration of 5µM as listed in Table 2.

#### 4.12.1. Measurement of FITC-Dextran Flux across Monolayers of Cultured Epithelial Cells

Caco2 cells were seeded on top of transwell filter chambers on 12-well plates (0.4 μm pore size; Falcon, Heidelberg, Germany) and EGCs were seeded below them (Figure 2A) as previously described [18]. After reaching confluence, cells were rinsed with PBS three times. Afterwards, the cells were incubated with fresh DMEM without phenol red (Sigma) containing 10 mg/mL FITC-dextran (4 kDa). Paracellular flux was assessed by taking 100 μL aliquots from the outer chamber over 2 h of incubation. Fluorescence was measured using a Tecan Microplate Reader (MTX Lab systems, Bradenton, USA) with excitation and emission at 485 and 535 nm, respectively. For all experimental conditions, permeability coefficients (P_E_) were calculated by the following formula (15): P_E_ = [(ΔCA/Δt) × VA]/S × ΔCL, where P_E_ = diffusive permeability (cm/s), ΔCA = change of FITC-dextran concentration, Δt = change of time, VA = volume of the abluminal medium, S = surface area, and ΔCL = constant luminal concentration.

#### 4.12.2. Depletion of GDNF form EGC Supernatants by Immunoprecipitation

GDNF was depleted from the EGC supernatant by using the *Immunoprecipitation Starter Pack* (GE Healthcare, Munich, Germany). Immunoglobulin sepharose bead slurries, Lysis buffer (150 mM NaCL, 1% IGEPAL CA-630, 0.5% sodium deoxycholate (DOC), 0.1% SDS, 50 mM Tris, pH 8.0) and wash buffer (50 mM Tris, pH8) were used as recommended by the manufacturer. EGC culture supernatants (12 mL) were harvested from confluent monolayers in T75-cell culture bottles. By means of 10 kDa Amicon Ultra-15 Centrifugal Filter Units (Merck Millipore, Darmstadt, Germany), the supernatant was concentrated to 500 µL by centrifugation for 25 min in a swinging bucket rotor centrifuge at 4.000 g. Then, 250 µL of the concentrated supernatant was incubated with 15 µL (0.2 mg/mL) GDNF antibody (R + D Systems, Abingdon, UK) for 2 h at 4 °C and mixed gently. Next, sepharose beads were added for one hour under the same conditions. After incubation, the sepharose beads were removed by centrifugation (12.000 g) and cell culture supernatants were used for experiments. The pelleted beads were washed 3 times with washing buffer and finally suspended in SDS lysis buffer containing 25 mmol/L HEPES, 2 mmol/L EDTA, 25 mmol/L NaF and 1% SDS.

#### 4.12.3. Generation of Cripsr/Cas9-Mediated Gene Knock-Down for GDNF in EGCs

The sgRNA for SpCas9-mediated genome editing was designed using the Chopchop-web based sgRNA design tool [43]. Two oligos targeting exon 3 of murine GDNF were designed and subcloned into two lentiviral vectors; sgRNA into pLentiCRISPR v2 (a gift from Feng Zhang, Addgene plasmid #52961) and sgRNA into pLKO5.sgRNA.EFS.GFP (was a gift from Benjamin Ebert, Addgene plasmid # 57822). Lentiviral particles were produced using HEk293T cells utilizing a second-generation lentiviral packaging system comprising *pPAX* and *pMD2*-packaging plasmids. EGCs in 6 well plates were infected 24 h later in the presence of polybrene with viral supernatant comprising both viral particles LKO-sgRNA-EFS GFP and CrisprV2.

Three days post-infection, cells were cultivated with Puromycin (15 µg/mL) for 1 week with medium changes every two days. Puromycin-positive clones were FACS-sorted for GFP-expression and reseeded as single cells in a 24 well plate. Single clones were propagated and loss of GDNF was assessed in single clones by western blot and *qPCR* as listed in Table 2.

#### 4.12.4. Measurements of Transepithelial Electrical Resistance (TER)

Transepithelial electric resistance was used to measure the paracellular permeability of ions with ECIS 1600R (Applied BioPhysics)at a frequency of 400 Hz, as described in detail previously [44].

### 4.13. Quantification of Immunostaining

Quantification of immunostaining was carried out as described previously [37]. In brief, a 10-μm line was placed orthogonal to the cell border with the cell border representing the middle of this line. The fluorescence pixel intensity was measured using ImageJ, which resulted in a graph with a maximum peak in the middle of the curve if the cell junction was intact. Loss of staining intensity at the cell borders resulted in a flattening of the curve (Appendix A). For each sample, at least four randomly chosen junctions were measured by a blinded observer, followed by statistical analysis.

### 4.14. GDNF ELISA

GDNF ELISA was used to quantify the concentration of GDNF in supernatants of cells as well as cell and tissue lysates, according to the manufacturer’s protocol (Abnova murine KA3041). The assay is a sandwich ELISA where 96-well plates are coated with anti-GDNF monoclonal antibody that binds soluble GDNF.

### 4.15. Cell Viability Assay

The cell viability assay *CellTiter-Glo 2.0* (Promega, Mannheim, Germany) was used in EGC after treatment with cytokines as described by the manufacturer. In brief, EGC were grown to confluence on 96-well plates. After serum starvation, cells were incubated with TNFα, LPS and LTA for 24 h. Staurosporine (Sigma Aldrich, St. Louis, USA) was used at a concentration of 500 ng/mL. Luminescent measurements were performed by using a microplate reader (Tecan Microplate Reader, MTX Lab systems, Bradenton, USA).

### 4.16. Statistics

Values are expressed as mean ± SEM. The number of experimental replicates (n) is presented in the figure legends. Statistical analyses were performed using GraphPad Prism 6.0 d (GraphPad, La Jolla, USA). For parametric data, possible differences were assessed with Student’s t-test or ANOVA and Two-Way Anova followed by Bonferroni post-test depending on the experimental design. For non-parametric data, Kruskal–Wallis following Dunn post-test was used for comparison of more than two groups, Mann–Whitney U test or Wilcoxon signed-rank-test were used for the detection of significant differences between two groups. Statistical significances were assumed for *p* < 0.05.

## Figures and Tables

**Figure 1 ijms-22-01887-f001:**
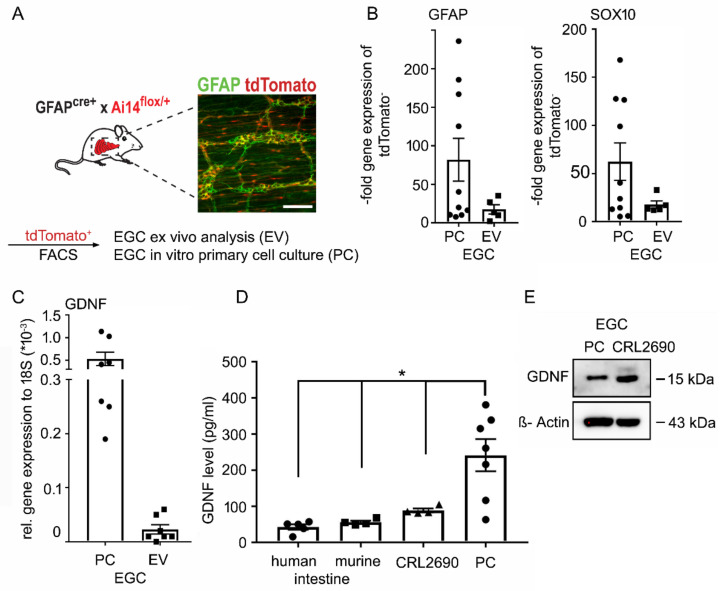
Enteric glial cells (EGC) express and secrete Glial cell line-derived neurotrophic factor (GDNF) at physiologically-relevant doses in vivo and in vitro. (**A**). Immunostaining of glial fibrillary acidic protein (GFAP) in the intestine from GFAP^cre^ x Ai14^floxed^ showed a co-localization of the inducible fluorescence protein tdTomato (red) and GFAP (green) at the plexus myentericus. (*n* = 8, scale 50µm). (**B**). *qPCR* of tdTomato positive cells that were sorted by Fluorescence-activated cell sorting (FACS) revealed that the cells express glial markers Sox10 and GFAP ex vivo and in primary cell culture (PC) (*n* = 5). (**C**). *qPCR* of that cells confirmed that tdTomato positive cells express GDNF ex vivo and in primary cell culture (*n* = 7). (**D**). GDNF ELISAs were performed of human (*n* = 5) and murine ileum lysates (*n* = 4) showing levels of GDNF at 56.0 ± 4.3 pg/mL and 42.6 ± 7.4 pg/mL. In supernatants from EGCs (CRL2690) (*n* = 4) GDNF levels of 88.7 ± 5.5 pg/mL and of primary EGCs GDNF levels of 241.7 ± 44.4 pg/mL were detected (*n* = 7; * = *p* < 0.05). (**E**). Lysates of primary EGCs as well as CRL2690 cells are positive for GDNF in Western Blots (*n* = 8). Quantification of the GDNF blots is shown in Appendix A.

**Figure 2 ijms-22-01887-f002:**
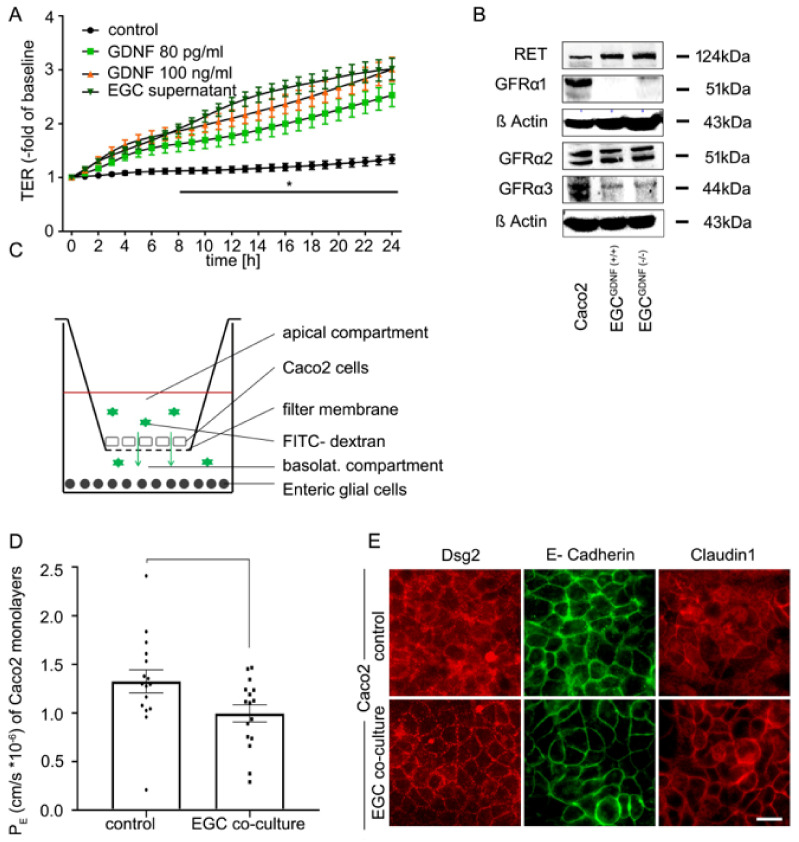
Co-culture of enteric glial cells (EGC) with Caco2 cells stabilizes epithelial barrier function. (**A**). Measurements of transepithelial electric resistance (TER) showed a significant increase beginning at 8 h after incubation with the supernatant of EGC and raised to 3.01 ± 0.1-fold of baseline (equates to 2.24-fold of controls) while incubation with glial cell line-derived neurotrophic factor (GDNF) at 100 ng/mL increased TER significantly to 2.99 ± 0.07 of baseline (2.23-fold of controls) after 24 h. Similar results were obtained when GDNF was applied at 80 pg/mL when TER increased to 2.52 ± 0.09 -fold of baseline (1.88-fold of controls); (*n* = 6, * = significant different vs. control after 8 h for all other conditions; p < 0.05, two-way ANOVA). Control = Caco2 monolayer without treatment. (**B**). Western Blots of Caco2 cells and EGC cells (wilde type (WT) and GDNF-knockdown) confirmed that all these cell lines express GDNF receptors and RET (*n* = 3). Quantification of the blots is shown in Appendix A (**C**). Schematic picture of the co-culture model of Caco2 cells and enteric glia cells. (**D**). Permeability coefficients (P_E_) of 4-kDa FITC dextran flux across Caco2 monolayers were significantly reduced 24 h after co-culture with EGC (*p* < 0.05, *n* = 16, student’s t-test). Control = Caco2 monolayer without treatment. (**E**). Immunostaining of the transwell filters showed an augmented and more linear staining pattern of desmosomal protein Desmoglein2 (Dsg2), adherens junction protein E-Cadherin and tight junction protein Claudin1 24 h after co-culture of confluent Caco2 cells with EGC compared to untreated controls (representatives are shown for *n* = 10, scale = 20 µm). Control = Caco2 monolayer without treatment.

**Figure 3 ijms-22-01887-f003:**
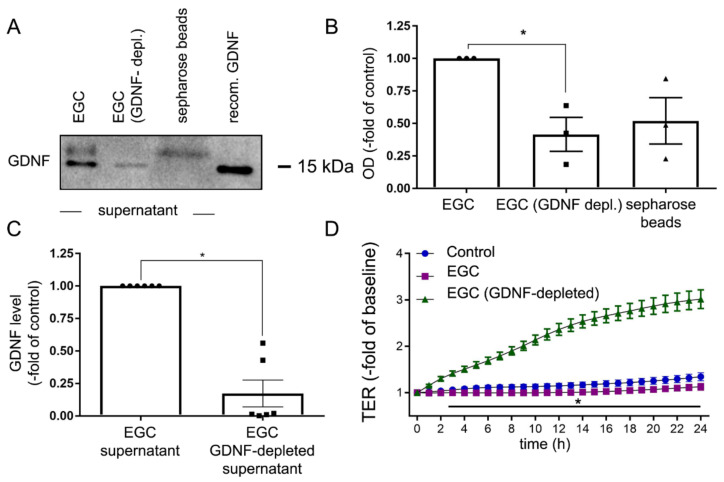
Depletion of glial cell line-derived neurotrophic factor (GDNF) attenuated effects of enteric glial cells (EGC) on enterocytes. (**A**). Representative Western blot is shown for GDNF with and without depletion of GDNF from EGC supernatants using sepharose beads and GDNF antibodies. Human recombinant GDNF served as a positive control (*n* = 3). (**B**). Quantification of the western blot signal showed a significant reduction of GDNF concentration in the cell lysates following the incubation with sepharose beads (*n* = 3, * = *p* < 0.05, Kruskal Wallis Test). Control = EGC supernatant. (**C**). ELISA- based measurements of the GDNF concentration in the supernatant showed a reduction of GDNF to 17 ± 10% after depletion with sepharose beads (*n* = 6, *p* < 0.05 Wilcoxon signed-ranked test). (**D**). Measurements of transepithelial electric resistance (TER) on Caco2 monolayers demonstrated, that the effect of EGC supernatant after depletion of GDNF by sepharose beads was blunted, compared to EGC supernatants without depletion of GDNF (*n* = 6, * = *p* < 0.05, two-way ANOVA). Control = Caco2 monolayer without treatment.

**Figure 4 ijms-22-01887-f004:**
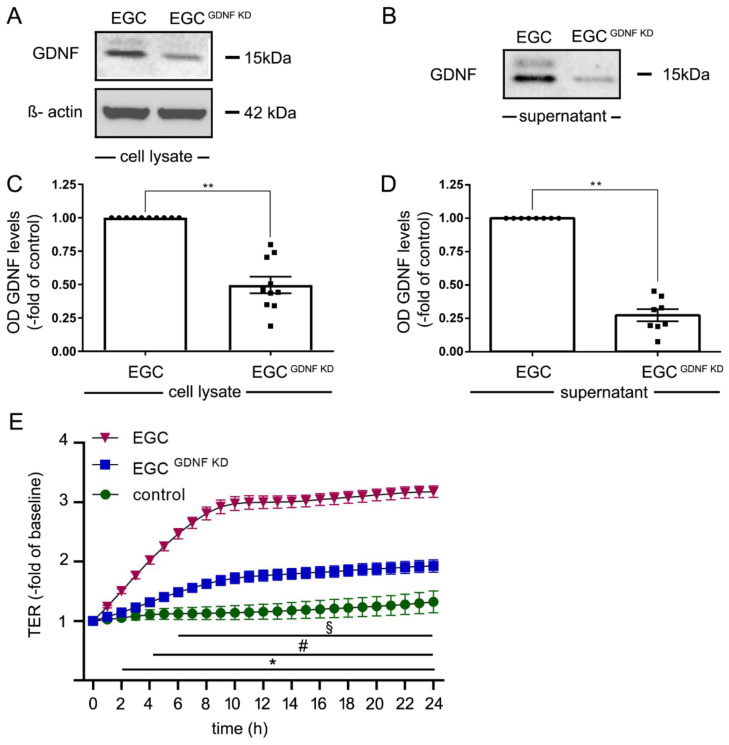
Knockdown of glial cell line-derived neurotrophic factor (GDNF) reduced effects of enteric glial cells (EGC) on barrier properties of Caco2 cells. (**A**). Representative Western blot of cell lysates of EGC to confirm reduced GDNF levels following knockdown of GDNF in EGC (EGC^GDNF KD^); (*n* = 10). (**B**). Representative Western blots of cell culture supernatants from EGCs and EGC^GDNF KD^ to confirm reduced secretion of EGCs into cell culture supernatants; (*n* = 8). (**C**). Quantifications of Western blots following knockdown of GDNF showed reduced levels to 45 ± 19% of control EGCs correlated to β-actin; (*n* = 10, ** = *p* < 0.01, Wilcoxon signed-ranked test). Control = EGC WT. (**D**). Quantifications of Western blots from supernatants from EGC supernatants compared to EGC^GDNF KD^ demonstrated reduced GDNF levels of 27 ± 12%; (*n* = 8, ** = *p* < 0.01, Wilcoxon signed-ranked test). Control = EGC WT. (**E**). Measurements of TER revealed that incubation of Caco2 with EGC supernatant significantly increased TER to 330% ± 16% after 24 h whereas incubation with supernatants from EGC^GDNF KD^ resulted in a significantly less pronounced increase of TER to 230% ± 18% (equates to 0.69-fold of EGC supernatants); (*n* = 10 for each condition, § = p< 0.05 control vs. EGC, # = *p* < 0.05 EGC vs. EGC^GDNF KD^, * = *p* < 0.05 control EGC; two-way ANOVA); Control = Caco2 monolayer without treatment.

**Figure 5 ijms-22-01887-f005:**
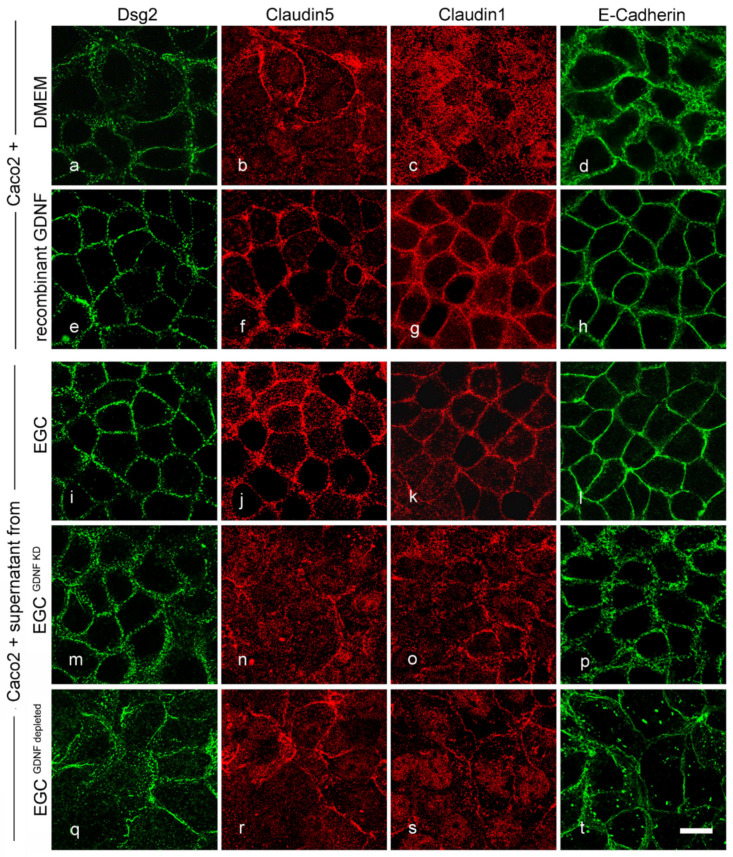
Enteric glial cells (EGC)-mediated effects on junctional proteins are glial cell line-derived neurotrophic factor (GDNF) dependent. Representative immunostainings under various conditions are shown. In immature Caco2 cells junctional proteins distribution of desmosomal Desmoglein2 (Dsg2) (**a**) and tight junction proteins Claudin5 (**b**), Claudin1 (**c**) and adherens protein E-Cadherin (**d**) were not regularly located at the cell borders. Application of recombinant GDNF augmented linear staining of all junctional proteins at the cell borders (**e**–**h**). Similar effects were observed following incubation with supernatants from EGCs (**i**–**l**). These effects of EGC on enterocytes were absent when cells were incubated with supernatants from EGC^GDNF KD^ (**m**–**p**) and with EGC supernatant in which GNDF had been depleted (**q**–**t**); (representatives are shown for *n* = 6, Scale 20 µm).

**Figure 6 ijms-22-01887-f006:**
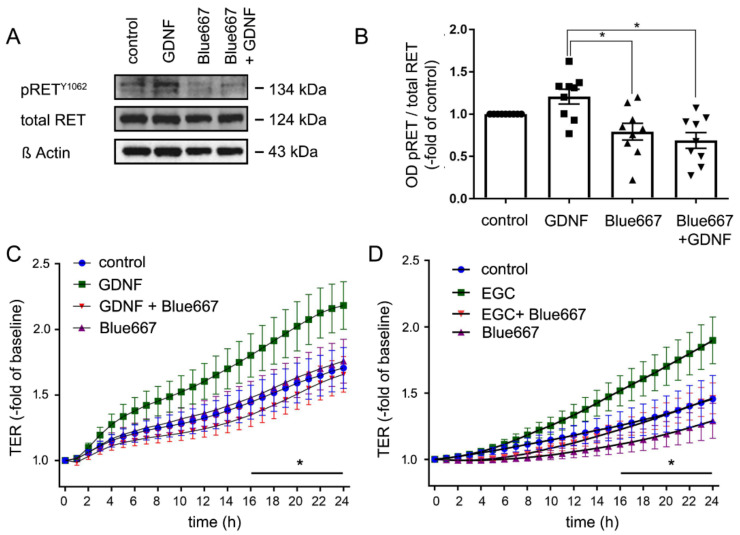
Effects of enteric glial cells (EGC) on epithelial barrier function are RET-dependent. (**A**). Representative Western blots of Caco2 cells are shown. Glial cell line-derived neurotrophic factor (GDNF) increased the ratio of phosphorylated RET^Y1062^ to 131 ± 9%- of controls, while application of Blue667 reduced basal phosphorylation of RET and blocked GDNF-induced RET phosphorylation to 67.5 ± 9% of controls; (*n* = 8). (**B**). Quantification of the western blots are shown which demonstrated that GDNF increased RET phosphorylation to 131.1 ± 9% of controls whereas Blue667 reduced RET phosphorylation; (*n* = 8, Kruskal Wallis Test); Control = Caco2 monolayer without treatment. (**C**,**D**). Transepithelial electric resistance (TER) measurements of Caco2 monolayers are shown. The inhibition of the RET by Blue667 blocked the effect of GDNF and of EGC supernatant on intestinal epithelial barrier maturation; (*n* = 8 for each set of experiments; * *p* < 0.05; 2-way ANOVA); Control = Caco2 monolayer without treatment.

**Figure 7 ijms-22-01887-f007:**
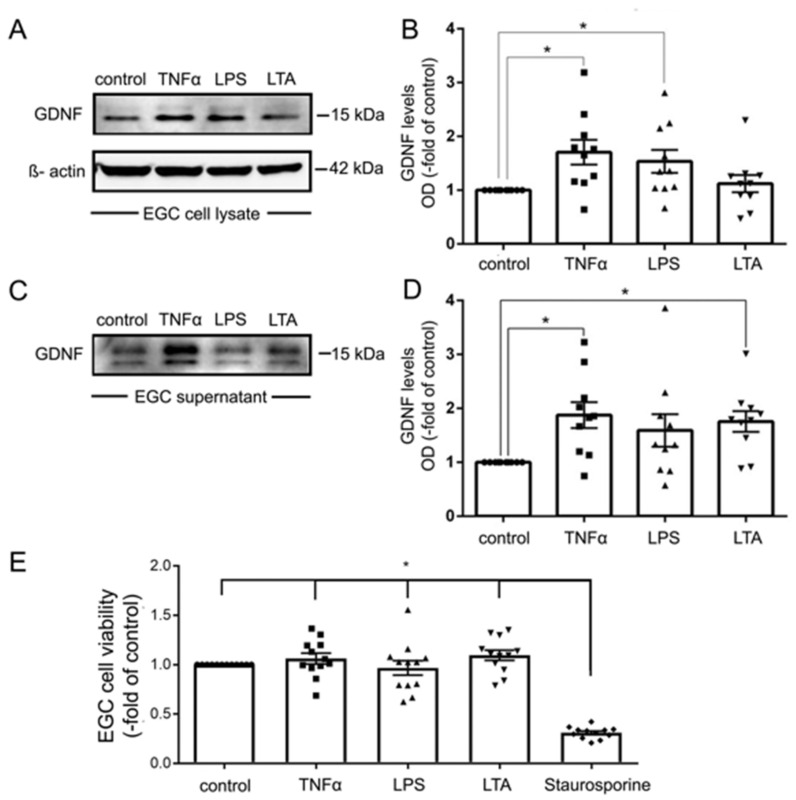
Tumor necrosis factor α (TNFα) increased the expression and secretion of glial cell line-derived neurotrophic factor (GDNF) in enteric glial cells (EGC). (**A**). Representative Western blot of EGC cell lysates showing that TNFα and lipopolysaccharide (LPS) but not lipoteichoic acid (LTA) led to increased expression of GDNF, membranes were reprobed for β-actin to ensure equal protein loading; (*n* = 9). (**B**). Quantification of all Western blots showed a significantly increased expression of GDNF by TNFα and LPS but not by LTA; (*n* = 9; * = *p* < 0.05, Kruskal Wallis Test); Control = Caco2 monolayer without treatment. (**C**). Representative Western blot of EGC cell culture supernatants is shown which demonstrates increased release of GDNF by TNFα and LTA but not LPS; (*n* = 9). (**D**). Quantification of all Western blots showed a significantly increased release of GDNF by TNFα and LTA but not by LPS (*n* = 9; * = *p* < 0.05, Kruskal Wallis Test); Control = Caco2 supernatant without treatment. (**E**). MTT assays served to test for cell viability. Except for incubation with staurosporine EGC cell viability was not changed for the different experimental conditions (*n* = 10; * = *p* < 0.05 control, TNFα, LPS, LTA vs. Strauosporine; Kruskal Wallis Test); Control = Caco2 monolayer without treatment.

**Figure 8 ijms-22-01887-f008:**
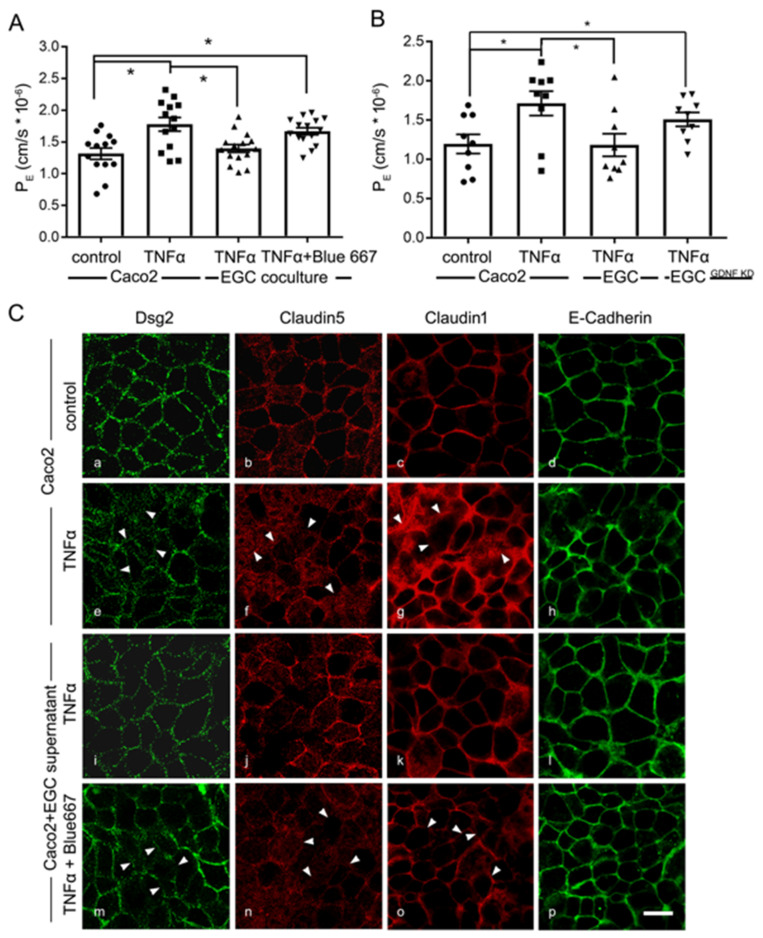
Barrier protective effects by enteric glial cells (EGC) in inflammation are glial cell line-derived neurotrophic factor (GDNF)-dependent. (**A**). Permeability coefficients (P_E_) of 4-kDa FITC dextran flux across Caco2 monolayers was increased following the application of Tumor necrosis factor α (TNFα) to 1.78 ± 0.10 × 10^−6^ compared to 1.32 ± 0.08 × 10^−6^. This was blocked under co-culture conditions together with EGC cells. Protective effects of the co-culture were diminished by Blue667 when P_E_ was 1.67 ± 0.06 × 10^−6^ (*n* = 9; * = *p* < 0.05; One-way ANOVA). Control = Caco2 monolayer without treatment. (**B**). In P_E_ measurements barrier protection of EGCs after TNFα incubation were absent when co-cultures were performed with EGC^GDNF KD^ (*n* = 9; * = *p* < 0.05; One-way ANOVA). (**C**). Representative immunostaining for junctional proteins Dsg2, E-Cadherin, Claudin1 and 5 are shown. TNFα reduced the staining pattern of these junctional proteins at the cell borders of Caco2 cells. This effect was attenuated when Caco2 cells were cultivated with EGC supernatants. The beneficial effects of EGC supernatants were blocked by RET Inhibitor Blue667; arrowheads point to examples of reduced or lost staining patterns at the cell borders (representatives are shown for *n* = 6, Scale 20 µm); Control = Caco2 monolayer without treatment.

**Figure 9 ijms-22-01887-f009:**
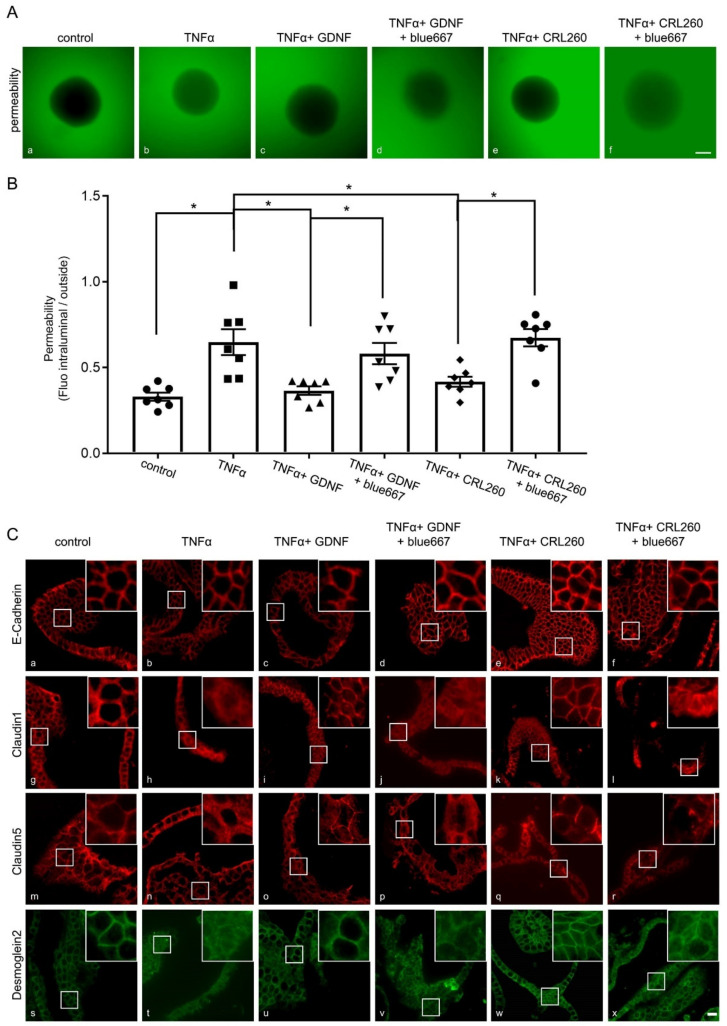
Effects of enteric glial cells (EGC) on organoids after the inflammation-induced breakdown of the intestinal barrier are dependent on RET-mediated pathways. (**A**): Representative images of organoids 1 h after incubation with 4 kDa FITC-Dextran are shown. Compared to controls (**a**) Tumor necrosis factor α (TNFα) increased the fluorescence inside the organoids (**b**), while simultaneous application of recombinant glial cell line-derived neurotrophic factor (GDNF) (**c**) or EGC supernatants attenuated (**e**) this inflammation-induced breakdown of the intestinal epithelial barrier, while RET inhibitor Blue667 blocked the effects of GDNF and EGC on organoid permeability (**d**,**f**). (representative figures are shown for *n* = 7, scale 120 µm); control = organoids without treatment. (**B**): Permeability in organoids was quantified by the quotient of the fluorescence in the organoid lumen and outside of the organoid. These measurements revealed an inflammation-induced increase to 1.94 ± 0.09-fold of controls. Incubation with GDNF or EGC supernatants reduced the effects of TNFα and the permeability quotient was reduced to 1.11 ± 0.08-fold of controls. This protective effect of EGC supernatants was blunted by concurrent application of Blue667, where the permeability quotient increased again to 2.04 ± 0.07-fold of controls (*n* = 7; * = *p* < 0.05; One-way ANOVA); control = organoids without treatment. (**C**): Representative immunostaining for junctional proteins Desmoglein2 (Dsg2), E-Cadherin, Claudin1 and 5 are shown. TNFα reduced the staining pattern of the tight junction proteins Claudin1 and 5 as well as the staining of Dsg2 at the cell borders (**b**,**h**,**n**,**t**). This effect was attenuated when organoid cells were cultivated with recombinant GDNF and EGC supernatants (**c**,**i**,**o**,**u** and **e**,**k**,**q**,**w**). The beneficial effects of EGC supernatants were blocked by RET Inhibitor Blue667 (**d**,**j**,**p**,**v** and **f**,**l**,**r**,**x**) (representatives are shown for *n* = 6, Scale bar = 80 µm); control = organoids without treatment (**a**,**g**,**m**,**s**).

**Table 1 ijms-22-01887-t001:** Primers used for *RT-PCR* of Sox10, GFAP and GDNF.

Primer Name	Sequence
mSox10_fw	GGACTACAAGTACCAACCTCGG
mSox10_rv	GGACTGCAGCTCTGTCTTTGG
mGFAP_fw	ACATCGAGATCGCCACCTAC
mGFAP_rv	CCTTCTGACACGGATTTGGT
mGDNF_fw	CAGTGACTCCAATATGCCTGA
mGDNF_rv	CCGCTTGTTTATCTGGTGAC

**Table 2 ijms-22-01887-t002:** Primers to verify genetic alterations by Crispr/Cas9.

Primer Name	Sequence
rat-ccGDNF 1f	CACCGTTCGAGAAGCGTCTTACCGG
rat-ccGDNF 1r	AAACCCGGTAAGACGCTTCTCGAAC
rat-ccGDNF 2f	CACCGTCACCAGATAAACAAGCGG
rat-ccGDNF 2r	AAACCCGCTTGTTTATCTGGTGAC
qPCR Primer	Sequence
rat-GDNF forward	AAGAGAGAGGAACCGGCAAG
rat-GDNF reverse	CGACCTTTCCCTCTGGAAT

## Data Availability

Data available in a publicly accessible repository. The data presented in this study are openly available in [repository name e.g., FigShare] at [doi], reference number [reference number].

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
