# Peer review of "Intestinal Epithelial Barrier Maturation by Enteric Glial Cells Is GDNF-Dependent"

_ijms, 2021, doi:10.3390/ijms22041887_

Round 1

Reviewer 1 Report

In the present study, the authors investigated the crosstalk between enteric glial cells (EGCs) and intestinal epithalial cells under physiological and inflammatory conditions, following application of TNF. In particular, Meir et al., focused their attention on the contribution of glial-derived neurotrophic factor (GDNF), secreted by EGCs, in intestinal epithelial barrier maturation. The authors reported that GDNF contributes to intestinal barrier maturation and inhibit inflammation-induced breakdown of the intestinal epithelial barrier by increasing the expression of tight junction proteins.

The manuscript is interesting but presents critical points:

MAJOR POINTS:

_ the authors need to carefully review the English style and the expressions. The authors are recommended to have a native english speaker spellcheck the manuscript thoroughly. Several concepts are not clearly discussed.

_ to better assess the effects of GDNF, secreted by EGCs, on tight junctions, the authors should perform a western blot analysis of E-cadherin, claudin-1 and Dsg2 in the following experimental conditions: Caco2 control, Caco2 treated with EGC supernatant and Caco2 treated with GDNF-depleted supernatant. This allow to better appreciate the effect of GDNF.

_ Supplementary figures: 5, 6, 7, 8 and 10 are missing. Please check the number of the suppl. Figures.

_ The authors reported that the effects of GDNF on enterocytes are mediated by RET-dependent signaling. Could the authors explain the intracellular signaling pathways coupled to the RET receptor activation in the effects observed?

_ The effect of GDNF, secreted by EGCs, on enteroids should be illustrated with high resolution images (Figure 9A). The current images are pretty, yet they do not clearly show fluorescence clearly. In addition, the authors should add the scale bar in Figure 9A.

_ As mentioned by the authors in the discussion, not only enteric glial cells, but also smooth muscle cells and enterocytes, express and secrete GDNF. Therefore, the authors should explain the rationale for comparing the levels of GDNF secreted by cells (rat CRL2690 cell line and primary mouse EGCs) with the levels of GDNF expressed in the ileum lysates. In addition, how do the authors justify the high level of GDNF secreted by primary mouse EGCs compared to that observed for CRL2690 cells and human and mice lysates?

MINOR POINTS:

_ Please add the statistical significance in Figure 2E.

_ Spaces and symbols are sometimes missing between number and unit (i.e lines 89, 117, 119, 182, 189, 208, 289 and 539).

_ The sentence page 8 line 225 is incomplete.

_ Please add the reference for the sentence page 15 line 400.

_ Please specify the cell line used in the “Materials and Methods” section line 584 page 19.

_ Line 603 page 20, indicate the test reagents used.

_ Lines 615 and 625, insert bracket for the reference 15 and 35.

_ Figure 1E, please indicate if there is a statistical significance between PC and other groups. 

_ Please replace the wording “TNF-a” with “TNF”.

Author Response

Dear editor,

Enclosed please find our revised manuscript which is entitled “Intestinal epithelial barrier maturation by enteric glial cells is GDNF-dependent”. All of the reviewers appreciated our manuscript but raised some concerns. According to the reviewers’ suggestions we have now carefully revised the whole manuscript and added new data as requested. Especially, as has been proposed by all reviewers a native English speaker carefully revised the English style and expressions. All comments raised by the reviewers were addressed and we have added a point-to-point reply to answer all questions.

In its revised form, we believe that the manuscript is substantially improved and we are now confident to meet the requirements that our manuscript will be accepted in the International Journal of Molecular Sciences. 

Yours sincerely

Michael Meir and Nicolas Schlegel

Point-to-point reply.

Reviewer 1:

MAJOR POINTS:

  1. The authors need to carefully review the English style and the expressions. The authors are recommended to have a native English speaker spellcheck the manuscript thoroughly. Several concepts are not clearly discussed.

According to the reviewer’s suggestion a native speaker revised the whole manuscript and carefully corrected the English style and expression. Additionally we have revised the discussion to ensure that all concepts are clearly discussed now.

  1. To better assess the effects of GDNF, secreted by EGCs, on tight junctions, the authors should perform a western blot analysis of E-cadherin, claudin-1 and Dsg2 in the following experimental conditions: Caco2 control, Caco2 treated with EGC supernatant and Caco2 treated with GDNF-depleted supernatant. This allows to better appreciate the effect of GDNF.

As suggested by this reviewer we have performed Western blot analyses for the junctional proteins E-cadherin, Claudin-1 and Dsg2 under the different condition. No significant differences in the total protein levels of the respective junctional proteins were observed in these experiments. This is line with our previous findings where we described that GDNF-mediated effects are not related to changes to protein expression but rather on recruiting of the junctional proteins to the cell borders (Meir et al Am J Physiol Gastrointest Liver Physiol . 2015 Oct 15;309(8):G613-24. doi: 10.1152/ajpgi.00357.2014; Meir et al. J Clin Invest. 2019 Jun 17;129(7):2824-2840. doi: 10.1172/JCI120261).

We added the new data a new supplementary Figure S3 and addressed this point in the results and in the discussion.

  1. Supplementary figures: 5, 6, 7, 8 and 10 are missing. Please check the number of the suppl. Figures.

The manuscript has been changed. The number of supplementary Figures have now been corrected.

  1. The authors reported that the effects of GDNF on enterocytes are mediated by RET-dependent signaling. Could the authors explain the intracellular signaling pathways coupled to the RET receptor activation in the effects observed?

The exact mechanisms following RET activation are not fully understood and a role for RET-dependent signaling in enterocytes is a relatively new observation. In general, the mechanism proposed by other groups for RET-dependent signaling involve signaling pathways that have been identified in the development of impaired intestinal epithelial barrier function in inflammatory bowel disease.   We have include this aspect in the Discussion in line 452 – 458

  1. The effect of GDNF, secreted by EGCs, on enteroids should be illustrated with high resolution images (Figure 9A). The current images are pretty, yet they do not clearly show fluorescence clearly. In addition, the authors should add the scale bar in Figure 9A.

We are glad that the reviewer appreciated our presentation of the intestinal organoids in Figure 9A. We understand the point raised by the reviewer to provide high resolution images. However, due to the fact that these photos were taken from the organoids within the Matrigel to document the distribution of 4 kDa FITC dextran it was not possible to avoid the slightly blurry appearance of the picture although the pictures were taken by a high-resolution camera. Nonetheless, a scale bar was included

  1. As mentioned by the authors in the discussion, not only enteric glial cells, but also smooth muscle cells and enterocytes, express and secrete GDNF. Therefore, the authors should explain the rationale for comparing the levels of GDNF secreted by cells (rat CRL2690 cell line and primary mouse EGCs) with the levels of GDNF expressed in the ileum lysates. In addition, how do the authors justify the high level of GDNF secreted by primary mouse EGCs compared to that observed for CRL2690 cells and human and mice lysates?

The rationale for comparing the levels secreted by cells with the levels of GDNF expressed in the ileum lysates was simply to double-check that the overall levels of GDNF we are working with were in a physiological range. It is clear that we cannot draw conclusions from these data which levels of GDNF derive from EGCs and other cell types within the whole tissue lysates from the terminal ileum. We have clarified this point in the results part and in the discussion.

The other question how we explain the high levels of GDNF secreted by the primary mouse EGCs compared to that for CRL2690 can in our opinion be explained by the fact that cell culture supernatants were not adjusted to the number of cells within the cell culture flasks so that different amounts of cells secreting GDNF may have induced this difference. This is speculative but this argument is supported by the Western blot analyses we performed for GDNF in the primary cell line in comparison with the CRL2690 cells: When we compare for GDNF expression using equal total protein amounts there is hardly a difference. We have pointed this our in the results section where we described the Western Blots for GDNF in EGCs.

MINOR POINTS:

  1. Please add the statistical significance in Figure 2E.

We added the statistical significance in the Figure 2E

  1. Spaces and symbols are sometimes missing between number and unit (i.e lines 89, 117, 119, 182, 189, 208, 289 and 539).

We carefully double checked the manuscript and corrected all mistakes regarding spaces and symbols

  1. The sentence page 8 line 225 is incomplete.

The sentence has been completed

  1. Please add the reference for the sentence page 15 line 400.

The reference [24] (Soret, R et al Neurogastroenterol Motil 2013, 25, e755-764.) that demonstrated the similarity of EGC through different species has been added

  1. Please specify the cell line used in the “Materials and Methods” section line 584 page 19.

The cell line was specified and is now described in the text.

  1. Line 603 page 20, indicate the test reagents used.

We added the sentence “as indicated in Figure 9” to address this suggestion by the reviewer

  1. Lines 615 and 625, insert bracket for the reference 15 and 35.

The references have been corrected as suggested by the reviewer

  1. Figure 1E, please indicate if there is a statistical significance between PC and other groups. 

The statistical significance has been added.

  1. Please replace the wording “TNF-a” with “TNF”.

Since lymphotoxin-alpha is sometimes described as TNFβ, we regard the wording TNFα more specific than just TNF. We therefore decided not to change the wording TNFα to TNF.

Reviewer 2

MAJOR POINTS:

  1. The authors reported that GDNF, secreted by EGCs, stabilizes intestinal epithelial barrier function. In this context, the expression of TJs should be evaluated also by western blot. Therefore, the authors should perform a western blot analysis of E-cadherin, claudin-1/5 and Dsg2 in the same experimental conditions of the Figure 5.

This comment is similar to the second point raised also by Reviewer 1. As described above in detail we added the new data and explained the results in the text.

  1. The authors investigated the protective effect of GDNF on TNF-induced impairment of the intestinal epithelial barrier. In Figure 8A and 8B, I suggest adding the following experimental condition: Caco2 cells (alone and in co-culture) treated with TNF in the presence of exogenous GDNF. This allows to better appreciate the protective effect of GDNF on intestinal epithelial barrier integrity.

We already addressed these protective effects of GDNF on TNFα-induced impairment of the intestinal epithelial barrier in a previous publication (Meir et al.; Neurotrophic factor GDNF regulates intestinal barrier function in inflammatory bowel disease; Journal of Clinical Investigation; DOI 10.1172/JCI120261). Since this would not add novel aspects we decided not  to repeat these experiments.

  1. The authors should justify the choice of the concentration used for the LTA-SA and report the articles present in literature on which they are based.

We used the concentration that was proposed by the manufacturer and cited previous publications that used LTA to stimulate Glia cells in the manuscript.

 MINOR POINTS:

  1. Lines 276 and 615, please uniform the name of lipoteichoic acid (LTA-SA or LTA ?).

We thank the reviewer for this suggestion and uniformed the expression for lipoteichoic acid to LTA throughout the text

  1. ­The authors should add the abbreviations in all figure legends to help the reader (i.e. Figure 1: GFAP, PC, EV, EGC…; Figure 2: Dsg2, EGC, GDNF..).

We agree with the reviewer and therefore explained all abbreviations during their first appearance in the figure legends

  1. Line 144 page 5, the full stop is missing.

We  added the full stop in line 144.

  1. The sentence page 8 line 225 is incomplete.

The sentence has been completed according to the reviewer’s suggestion

  1. Please explain the meaning of the abbreviation “ME” page 18 line 560.

The abbreviation of muscularis externa as ME has been added at its first appearance

  1. Lines 615 and 625, insert bracket for the reference 15 and 35.

We added the brackets for the references and thank the reviewer for that notice

  1. Please use the italics for the following sections: 4.12.1 (line 663); 4.12.2 (line 676); 4.12.3 (line 692) and 4.12.4 (line 708).

The headings are written in italics now

  1. Please increase the dimension of the symbols for the Figure 3D, 4E, 6C and 6D.

We agree with the reviewer that in the original figures the symbols of the graphs were hard to discriminate. To improve this, we increased the dimension of the symbols and used different colours.

  1. Spaces and symbols are sometimes missing between number and unit. Please, check the manuscript.

We carefully double checked the manuscript and corrected all mistakes regarding spaces and symbols

Reviewer 3

  1. Have the authors analysed the cell viability after FACS in experiments included in Figure 1? It would be great if authors confirm that the viability of the cells was not affected.

We thank the reviewer for this comment. As stated in the materials and methods section, only living cells were sorted by FACS. Those cells could be easily expanded and cultivated afterwards. Since this fact was not made clear in the first place, we explained the method of sorting living cells by Hoechst staining in the materials and methods section line 575 and 602.

  1. How authors explained in Figure 1A that the TER is higher with the EGC supernatant than the GDNF 100ng/ml since they have seen that the GDNF levels of the EGC supernatant are around 88,7 ng/ml?

We appreciate this comment by the reviewer. There is indeed a tendency that EGC supernatant leads to a higher TER than that following the application of recombinant GDNF. This difference however was not significant. One explanation for this might be the presence of additional factors secreted by EGC. We have pointed out  the lack of a significant difference in the different conditions out clearer in the text.  

  1. For all the Western Blots included in the manuscript, a quantification of all the n described in the Figure Legends should be included. In addition, the full-length blots are not included in the supplementary material. Please add such pictures and show the band of the molecular weight marker in each membrane.

We totally agree with the reviewer, that full length blots should be included in the manuscript. To our knowledge we already included a pdf file showing all blots in our previous submission. We also agree that quantification of Western blots should be included. The missing quantification of blots in figure 1 and 2 are now shown in a new supplementary figure 1, the quantification of the blots in figure 3A can be found in Figure 3B. The analysis of the optical density of 4A and B can be found in Figure 4C and D. The quantification of 6A is shown in Figure 6B. Analysis of Blots in Figure 7A, C are demonstrated in Figure 7B, D.

  1. The Figure Legend 1 is wrong. Check letters D and E.

We  corrected the Figure legend 1

  1. Check line 189. The numbers of the reduction of GDNF are wrong.

We thank the reviewer for that comment and corrected the reduction of GDNF

  1. The references 15 and 17 are the same.

We are grateful for this comment and reviewed the references.

Reviewer 2 Report

Meir M et al. investigated the role of GDNF secreted by enteric glial cells (EGCs) in the intestinal epithelial barrier. The topic is of interest since the alteration of epithelial barrier is a feature of IBD patients. Authors have performed numerous in vitro experiments where they report that specifically the GDNF secreted by EGCs is involved in the intestinal epithelial barrier function. Among the manuscript, authors have demonstrated by several approaches (knock-down by CRISPR/Cas system, pharmacological blockage of the receptor, antagonism with antibody and exogenous administration of the agonist) their main conclusion. In addition, most of the experiments are well designed and the scientific quality of the manuscript is very high. Although this reviewer misses some clinical results in IBD patients or in vivo experiments with mice, the same group has reported such information in their previous manuscripts. Nevertheless, there are some minor concerns that need to be addressed for this referee:

  • Have the authors analysed the cell viability after FACS in experiments included in Figure 1? It would be great if authors confirm that the viability of the cells was not affected.
  • How authors explained in Figure 1A that the TER is higher with the EGC supernatant than the GDNF 100ng/ml since they have seen that the GDNF levels of the EGC supernatant are around 88,7 ng/ml?
  • For all the Western Blots included in the manuscript, a quantification of all the n described in the Figure Legends should be included. In addition, the full-length blots are not included in the supplementary material. Please add such pictures and show the band of the molecular weight marker in each membrane.
  • The Figure Legend 1 is wrong. Check letters D and E.
  • Check line 189. The numbers of the reduction of GDNF are wrong.
  • The references 15 and 17 are the same.

Author Response

(The authors gave the same response as above.)

Reviewer 3 Report

In the present study, Meir et al., investigated the role of GDNF, secreted by enteric glial cells (EGCs), in intestinal epithelial barrier maturation. The authors, besides to examine the GDNF levels in EGCs (CRL2690 and primary cells) and in human and murine intestinal lystates, investigated the role of GDNF on intestinal epithelial barrier maturation under physiological and inflammatory condition, following application of TNF. They observed that GDNF improves intestinal epithelial barrier maturation by regulating the expression of tight junction proteins, such as E-cadherin, claudin-1 and desmosomal cadherin Dsg2. These effects were blunted when the experiments were carried out using EGC GDNF knock-down cells or by RET receptor blockage.

The manuscript is interesting but presents critical points:

MAJOR POINTS:

_ the authors need to carefully review the English style and the expressions.

_ The authors reported that GDNF, secreted by EGCs, stabilizes intestinal epithelial barrier function. In this context, the expression of TJs should be evaluated also by western blot. Therefore, the authors should perform a western blot analysis of E-cadherin, claudin-1/5 and Dsg2 in the same experimental conditions of the Figure 5.

_ The authors investigated the protective effect of GDNF on TNF-induced impairment of the intestinal epithelial barrier. In Figure 8A and 8B, I suggest adding the following experimental condition: Caco2 cells (alone and in co-culture) treated with TNF in the presence of exogenous GDNF. This allow to better appreciate the protective effect of GDNF on intestinal epithelial barrier integrity.

__ The authors should justify the choice of the concentration used for the LTA-SA and report the articles present in literature on which they are based.

MINOR POINTS:

_ Lines 276 and 615, please uniform the name of lipoteichoic acid (LTA-SA or LTA ?).

­_ The authors should add the abbreviations in all figure legends to help the reader (i.e. Figure 1: GFAP, PC, EV, EGC…; Figure 2: Dsg2, EGC, GDNF..).

_ Line 144 page 5, the full stop is missing.

_ The sentence page 8 line 225 is incomplete.

_ Please explain the meaning of the abbreviation “ME” page 18 line 560.

_ Lines 615 and 625, insert bracket for the reference 15 and 35.

_ Please use the italics for the following sections: 4.12.1 (line 663); 4.12.2 (line 676); 4.12.3 (line 692) and 4.12.4 (line 708).

_ Please increase the dimension of the symbols for the Figure 3D, 4E, 6C and 6D.

_ Spaces and symbols are sometimes missing between number and unit. Please, check the manuscript.

Author Response

(The authors gave the same response as above.)

Round 2

Reviewer 1 Report

no comments to add

Author Response

We thank the reviewer for his time and his effort.

Reviewer 3 Report

Major point:

_ supplementary Figure S3 should show the expression of the TJs by western blot analysis. There is a mistake in the number of suppl. Figures. The data regarding the expression levels of TJs are missing. Therefore, the authors should check for missing data and introduce them in the revised version of the manuscript

Minor point:

_ please check the sentence line 115 page 2. Delete in word "confirmed"

Author Response

We thank the reviewer for his time and his effort dedicated to providing feedback on our manuscript. We changed the manuscript according to the reviewer's suggestion.

In its revised form, we believe that the manuscript is substantially improved and we are now confident to meet the requirements that our manuscript will be accepted in the International Journal of Molecular Sciences. 

Yours sincerely

Michael Meir and Nicolas Schlegel

Major point:

_ supplementary Figure S3 should show the expression of the TJs by western blot analysis. There is a mistake in the number of suppl. Figures. The data regarding the expression levels of TJs are missing. Therefore, the authors should check for missing data and introduce them in the revised version of the manuscript

In the previous version the quantification of tight junction proteins Claudin1 and Claudin5 was already shown in suppl. figure S3. However, to make this clearer we added headings over each graph to make it easier for the reader to appreciate the quantification.

Furthermore, we agree with the reviewer, that the information of suppl. Figure S3 should be added to the main text. Therefore we added line 230 -237 to the manuscript and corrected the number in the suppl. Figures.

Minor point:

_ please check the sentence line 115 page 2. Delete in word "confirmed"

We thank the reviewer for this comment and deleted the word confirmed.